# "Stepping Up": A Decade of Relationship Violence Prevention

**Catherine J. Carter-Snell** [1,*] **and D. Gaye Warthe** [2]

1  School of Nursing and Midwifery, Mount Royal University, 4825 Mount Royal Gate SW, Calgary, AB T3E 6K6, Canada
2  Health Community and Education, Mount Royal University, 4825 Mount Royal Gate SW, Calgary, AB T3E 6K6, Canada; gwarthe@mtroyal.ca
*  Correspondence: ccartersnell@mtroyal.ca

**Abstract:** Students in postsecondary education are at high risk for experiencing relationship violence, including dating, domestic, and sexual violence. This can result in significant mental and physical health consequences. A relationship violence prevention program has been offered and evaluated for over 10 years at a Canadian university. It is based on a social–ecological model of violence prevention and best practices. Students who completed both pre- and post-program evaluations were used as their own controls to evaluate the effects of the program. Significant changes were noted for most aspects of the program in knowledge, attitudes, and behavioural intents each year, and these changes persisted for up to six months on most measures. The sample sizes were small and potentially overestimated the effect of the program if results were reported individually. Meta-analysis was used to pool the data and examine the effects of the program across the decade. The results indicated that the program was effective in changing knowledge, attitudes, and behavioural intents immediately following the program, but there were insufficient paired data to conduct six-month meta-analyses. Suggestions are made for future programs and further research.

**Keywords:** dating violence; intimate partner violence; healthy relationships; prevention; relationship violence

## 1. Introduction

Young adults at North American post-secondary education institutions are at high risk for relationship violence (Cantor et al. 2020). Most students are in dating relationships, but some may be married or cohabiting. The term "relationship violence" is used here to include both dating and domestic violence, both of which may also include sexual violence. Relationship violence is more common in dating situations than in spousal relationships (Burczycka and Conroy 2018; Conroy 2021). Sexual assaults by intimate partners (current or past) account for 20% of all sexual assaults in Canada (Cotter and Savage 2019) and 30% in the United States (Bagwell-Gray et al. 2015). The risk of relationship violence is higher if students have witnessed family violence as children or adolescents (Forke et al. 2018) or have higher rates of other types of adverse childhood experiences. Women and gender-diverse individuals are at high risk (Brewerton et al. 2021), as are cultural minorities (Fagan 2022). The long-term consequences of violence are well-recognised to include physical and psychological trauma, chronic health issues, and impacts on academics, employment, and future relationships. The prevention of relationship violence is therefore critical.

### 1.1. Prevention Program Effectiveness

Prevention programs for relationship violence in post-secondary education are varied in their content and their effectiveness. Elements of an effective prevention program typically include repeated exposures and sufficient time spent (indicating a dose relationship to prevention) as well as a focus on skills that support healthy relationships and bystander efficacy strategies, which include some type of follow-up event (Finnie et al. 2022; Crooks et al.

2019; Kovalenko et al. 2022; Wong et al. 2021). An environmental scan of 85 post-secondary campuses across Canada was conducted to identify programs and policies related to relationship violence (Warthe et al. 2017, 2018). The study methods, which included reviewing publicly available information and unpublished literature and conducting interviews, identified that the most common types of programming were bystander interventions, followed dedicated sexual violence resources, consent, and, increasingly in 2018, a focus on men as allies in prevention. An emphasis on sexual violence prevention was seen, with 85% of schools having some type of policy on sexual violence or misconduct. Missing from most schools were policy and programming on dating relationships and healthy relationships. A systematic review of college relationship violence programs identified significant changes in knowledge and attitudes as a result of the programs but found that the programs were less effective at consistently impacting bystander behaviours (Finnie et al. 2022; Wong et al. 2021). Their findings supported the previously identified need for multiple exposures, as well as the need for content to be engaging and delivered in person.

### 1.2. Background to Current Study

The investigators used programming principles to design a program in 2010 called "Stepping Up" (Kostouros et al. 2016). This program was a modification of a Canadian high school program called "Making Waves", which consisted of a student-led weekend retreat for students to discuss issues related to dating violence (Cameron et al. 2007). In the Making Waves program, students discussed topics in three major categories: boundaries and communications; healthy relationships; and gender and media effects on relationships. Students led the discussions but were given content to discuss and, later, assess.

The Stepping Up program was based on a social–ecological approach to violence prevention (Centers for Disease Control 2009), with a focus on individual, relationship, and community aspects of prevention. This included involving relevant community agencies, or community partners, in the planning and delivery of the program to establish awareness of resources and build connections between students and community partners. Modifications consisted primarily of more flexible peer-led delivery and some additions to the modules. The healthy relationships module also had to contain some discussion of unhealthy relationships and communication. Since communication and boundaries were inherent in the content of the healthy relationships module, the boundaries and communication module was shifted to focus intentionally on bystander intervention skills (Kostouros et al. 2016). A fourth module on sexual relationships was added, ensuring that a discussion was included of healthy as well as unhealthy/violent sexual relationships and consent. Students recruited to be peer facilitators worked together with faculty, program staff, and community partners for three months before the program was delivered. Peer facilitators identified up to three topics in each of the four modules that were relevant to students at our university and then created innovative ways for students to explore these topics. Allowing the peer facilitators to choose the topics within the module was consistent with the adult learning principle of relevance to students (Knowles et al. 2015) and provided peer facilitators with a sense of commitment and ownership.

The modules were delivered over two to three days to up to 60 students in groups of 15 maximum. Knowledge, attitudes, and behavioural intents were measured before, immediately after, and approximately six months after the program. Anonymous participant codes allowed us to collate each survey and use students as their own controls. We identified statistically significant differences each year in participants' self-rated knowledge changes as well as in most areas of content assessed. Most of these changes in knowledge, attitudes, and behavioural intent persisted for six months and, in some cases, became even stronger. The most notable limitation of the single studies was the attrition in completing surveys between the pre-measurement and the immediate post-measurement. Only 20–30% of the participants completed the post-surveys, and they did not always complete each survey. This resulted in some years with a size of less than 10 pairs in the post-sample. Although paired samples reduce the required sample size, it was still too minimal to allow meaningful

interpretation of each study. As a result, the single-study data have not previously been published.

Meta-analysis of small sample replication studies is one technique that can be used, particularly if there is reason to believe that the effect sizes and confidence intervals are similar between studies (Zondervan-Zwijnenburg and Rijhouwer 2020). The purpose of this study was to use meta-analytic techniques to explore the pooling of data for the in-person programs delivered pre-COVID-19. A focus on the pre-COVID-19 program allowed the authors to more confidently identify the estimated effect of the program. Specific questions to address were estimates of the differences in knowledge, behaviour, and attitudes resulting from the program across the years.

## 2. Methods

### 2.1. Studies and Participants

The program was supervised by researchers who acted as faculty advisors, along with at least one other faculty advisor and a program coordinator. The peer facilitator group chose topics within each module that were relevant. There were some general guidelines: the peer-led activities were required to be engaging and, although they often included discussions, they also included role-playing, video reviews, media analysis, and games. Community partners and faculty advisors worked with the peer facilitators to develop the module activities to ensure that they were not unnecessarily triggering, that they were manageable, and that any content provided was correct. Although the peer facilitators chose topics and activities within each module, the content was surprisingly similar each year (e.g., consent in sexual relationships, communications in healthy relationships, how to intervene in boundaries/bystander modules, the role of gender in violence, and relationships in gender/media). Many of the peer facilitators returned the next year or changed from participant to peer facilitator. This provided some consistency and continuity in the delivery of the program.

These four modules were delivered over two to three days to up to 60 participants in four groups of 15. Coffee breaks and lunch breaks were included to allow for informal discussion time. Peer facilitators focused on participant engagement with the activities. Community partners attended as experts and resources for each module in case specific content or questions arose. A follow-up community awareness activity was expected from the participants approximately a month or two after the program ended. The program participants subsequently held a community fair on campus and invited community partners to set up booths. Students arranged displays, games, photo booths, and giveaways to engage other students in conversations about healthy relationships and bystander intervention.

The first phase each fall was recruiting peer facilitators to identify and create activities for each module. They met at least monthly with the researchers/faculty advisors and the program coordinator, along with the community partners, to further refine the activities. In late fall, student participants were recruited via academic advisors, faculty, university emails, posters, web announcements, and word of mouth from peer facilitators. The second phase was the program delivery. Participants attended the program, which included the Friday session initially and was used to share a meal, meet with other participants, and introduce the topic of healthy and unhealthy relationships. Participants were assigned to their groups and completed at least one ice-breaker activity so that the participants could get to know their group members. Participants remained in the same groups on Saturday as they moved through each of the four modules. Breakfast and lunch were served on Saturday, providing further opportunities for conversation and sharing. Community partners also used these times to share information about their programs and resources. At the end of the modules, the students were gathered to remind them of their community awareness/prevention project and the date of the community fair. The early groups were given time on Sunday to start planning these activities, and later, groups arranged to meet at later times that suited their schedules. The Friday through Sunday format was used until 2014–2015 and was collapsed to a Saturday and Sunday model in 2015–2016,

reflecting the time demands of both the peer facilitators and participants. The ice-breakers and group assignments were moved to Saturday morning, so the content remained the same. Programs in 2010–2019 were in-person and included shared meals and breaks for informal discussion.

All in-person programs prior to the COVID-19 pandemic were included in the analysis, for a total of seven programs between 2010 and 2011 and the 2019–2020 academic years. Only the participant data were included in this study. Peer facilitator data were previously discussed in another publication (Kostouros et al. 2016). The programs offered during COVID-19 and the following year were not included in this analysis. The format of the program was significantly altered in duration to accommodate the virtual format (two evening sessions two weeks apart). For two years, programs were only offered virtually, and although the 2023 program was in-person, a shortened format was used. It was decided that there were too many unique variables in the pandemic/post-pandemic programs to allow meaningful comparison in a meta-analysis. For example, virtual participants were able to turn their cameras off and keep their participation anonymous. This may have affected interaction and program effectiveness. The short interactions and limited time together also potentially limited sharing and exposure. These three programs (two virtual and one post-pandemic in-person) were studied separately.

### 2.2. Ethics

Ethics approval was received from the university's Human Research Ethics Board. After receiving informed consent, participants were asked to choose a unique code using prompts each time so they could recreate the same code for subsequent questionnaires. In this way, their responses could be linked, but their identity remained anonymous.

### 2.3. Measurement Instruments

Prior to entering the program, students completed a Dating Relationship Scale, or DRS (Warthe 2011). There were 41 questions that, in addition to collecting demographics (age, sex. gender identity, and ethnicity), were focused on dating relationship status, experiences of relationship violence in one or more relationships, witnessing relationship violence in the home as a child (<12 years) or teenager, and wither participants had experienced other adverse childhood experiences as children or teens. The other adverse experiences included verbal, emotional, spiritual, financial, physical, and sexual abuse. The Likert-style questions used the following anchors: never, rarely, sometimes, or frequently.

Participants also completed a Knowledge, Attitudes, Behaviours, and Behavioural Intents (KABBI), which was adapted from the Making Waves program by adding questions related to sexual assault and healthy sexual relationship content or myths common in the literature. Behavioural intents were used as a proxy for actual behaviour. Intent is related to actual behaviour, especially if the intention and attitudes are strong (Bhattacherjee and Sanford 2009; Conner and Norman 2022). Participants were asked to complete the KABBI before the program and then again immediately after the program and after approximately six months, well after the community event was completed.

The first part of the KABBI (Appendix A) includes questions related to participants' self-rating of their level of knowledge in 13 areas. These included knowledge about healthy relationships, warning signs of abuse, interventions for dating violence, healthy sexual relationships, sexual assault myths, bystander interventions, personal strategies for prevention, dating violence community resources, sexual violence community resources, healthy communication skills, personal boundaries, gender stereotypes, and the role of media in relationships. They rated their knowledge from 1 (very little knowledge) to 5 (extremely well-informed).

The remainder of the questions on the KABBI pertained to specific questions relevant to each of the four modules: healthy relationships, sexual relationships, gender/media impact, and bystanders/boundaries. Participants were asked to respond to specific knowledge, attitude, and behavioural intent questions relevant to each section. They answered using a

five-point Likert scale with anchors of completely disagree (1) to completely agree (5). The KABBI was completed prior to the program, immediately afterwards, and at approximately six months post-program. The creators of the KABBI did not report validity or reliability data. They did use some questions that were worded in reverse (expecting a decrease in score vs. an increase) in order to assess internal consistency, but the results were not reported. All negative items on the KABBI were rescored in our study to show positive increments only—if their score increased, they had more positive attitudes or behavioural intents or increased knowledge. It has not been evaluated since we added the sexual relationship questions.

### 2.4. Data Analysis

Participants for each year were compared descriptively for any significant differences in demographic characteristics, using data from the Dating Relationship Scale collected at the beginning of each study (Table 1).

**Table 1.** Participants' Dating Relationship Scale.

|  | Year | 2011 | 2013 | 2015 | 2016 | 2017 | 2018 | 2019 |
|---|---|---|---|---|---|---|---|---|
| Characteristic | n | 29 | 39 | 47 | 53 | 32 | 24 | 20 |
| Sex % | Male | 33.3 | 12.8 | 20.4 | 11.1 | 16.1 | 4.4 | 5.0 |
|  | Female | 66.7 | 84.6 | 79.6 | 88.9 | 83.9 | 95.7 | 95.0 |
|  | Transgender/nonbinary | 0.0 | 2.6 | 0.0 | 0.0 | 0.0 | 0.0 | 0.0 |
| Gender Identity % | Heterosexual | 96.7 | 92.1 | 77.6 | 75.5 | 76.7 | 87.0 | 85.0 |
|  | Gay, lesbian, queer | 0.0 | 2.6 | 12.2 | 5.7 | 13.3 | 0.0 | 0.0 |
|  | Bi/pan/asexual | 3.3 | 5.3 | 10.2 | 18.9 | 10.0 | 13.1 | 15.0 |
| Age range | 18–24 | 44.8 | 43.6 | 63.8 | 73.6 | 93.8 | 83.3 | 60.0 |
|  | 25–29 | 31.0 | 15.4 | 19.1 | 13.2 | 6.3 | 8.3 | 30.0 |
|  | 30+ | 24.1 | 15.4 | 17.0 | 13.2 | 0.0 | 8.3 | 10.0 |
| Ethnicity/Race % | Indigenous | 6.7 | 7.7 | 6.1 | 7.4 | 12.5 | 12.5 | 0.0 |
|  | Caucasian | 63.3 | 82.1 | 67.4 | 66.7 | 59.4 | 58.3 | 70.0 |
|  | Other | 23.3 | 10.2 | 26.5 | 25.9 | 28.1 | 29.2 | 30.0 |
| Relationship now% | Single, never dated | 13.3 | 0.0 | 6.3 | 9.3 | 3.1 | 13.0 | 20.0 |
|  | Not dating | 46.7 | 35.9 | 39.6 | 42.6 | 40.6 | 13.0 | 30.0 |
|  | Currently dating | 6.7 | 35.6 | 22.9 | 29.6 | 53.1 | 56.5 | 30.0 |
|  | Cohabiting-married | 30.0 | 23.0 | 25.0 | 15.9 | 3.1 | 17.4 | 20.0 |
| Experienced RV% * | Verbal abuse | 48.2 | 64.9 | 60.9 | 51.0 | 56.7 | 47.8 | 47.4 |
|  | Emotional abuse | 48.1 | 65.8 | 76.1 | 60.8 | 76.7 | 60.9 | 57.9 |
|  | Spiritual abuse | 14.8 | 21.0 | 10.9 | 21.6 | 31.0 | 13.1 | 21.1 |
|  | Financial abuse | 29.6 | 23.7 | 29.6 | 19.6 | 26.7 | 17.4 | 15.8 |
|  | Physical abuse | 33.3 | 44.7 | 21.7 | 27.5 | 23.3 | 21.7 | 21.1 |
|  | Sexual Abuse | 25.9 | 29.0 | 28.3 | 16.8 | 43.3 | 30.4 | 26.3 |
| Witnessed RV at home * | As child | 44.8 | 57.9 | 54.4 | 46.2 | 60.0 | 54.2 | 60.0 |
|  | As teen | 27.5 | 62.2 | 58.7 | 47.1 | 53.3 | 52.2 | 65.0 |

* = sometimes or often/frequent.

The data for each year were also compared descriptively for each area of knowledge, attitudes, and behavioural intent. The questions pertaining to each aspect (knowledge, attitudes, or behavioural intents) across the four modules were added together. A higher score indicated greater agreement with the statements and an increased understanding of abusive versus healthy situations. The scores from the time immediately post-program and after six months were compared with their baseline scores. *t*-Tests were used for the knowledge self-rating scores. Final difference scores were calculated for the categories of knowledge, attitudes, and behavioural intents.

Meta-analysis was used to estimate the effect of the program using each year's results. This is a useful tool when individual studies are too small to detect a difference or when

the effect may be overestimated due to the size (Turner et al. 2013). A mean difference approach was used for analysis (Cochrane Collaboration 2018). A fixed-effects model was also used, as it was reasonable to assume that the studies were similar enough to have a common effect (Tufanaru et al. 2015). The same format, researchers, and measurements were used throughout, and the four topic areas remained constant. The risk of bias for non-randomised studies can be an issue in meta-analysis. The risk of bias in these studies was moderate, mainly due to selection bias. Only a small number of participants completed the repeated measurements to allow the assessment of change (Sterne et al. 2016, 2020).

Linear regression analysis was used to calculate the relative contribution of the questions to the change score. The final knowledge, attitude, or behavioural intent scores were used as the dependent variable, and the scores on each question included in that area were used to examine relative contributions to the final score.

## 3. Results

### 3.1. Participants

There was considerable attrition in responding to the immediate post-program KABBI and the six-month response rates. Approximately 30% responded to the post-program survey and 20–25% to the six-month survey despite a reminder email to all. This rate, however, is consistent with most electronic surveys. The actual sample used in the analysis was less than that, however, as only paired data were used to help reduce heterogeneity. In 2018, the six-month data were not used, as there was only one pair, thus limiting analysis, and in 2019, the six-month survey was not sent out due to a change in coordinators and miscommunication.

The characteristics of the participants in the initial samples shown are based on data from the Dating Relationship Scale (Table 1). The sample size shown for each year is less than the number of actual participants, as not all completed the DRS (or the KABBI) before the program. As shown, while the participants were mainly female, Caucasian, and heterosexual, there was some diversity across programs and within programs. The majority were under 24 years of age, but there were numerous mature students as well. Not all were in relationships, but those who were described these mostly as committed relationships. In later years, an increasing percentage who had never dated attended.

There was a relatively high percentage of participants who reported witnessing relationship violence at home, either as a child or as a teen. Prevalence data are difficult to determine for this element, as rates of family violence differ across locations, as do definitions. An American study suggested that 20 to 40% of adults report having witnessed family violence (Davis and Briggs 2000). The DRS data for those who completed the post KABBIs was similar to the large-group DRS data with minor exceptions—the few who completed the 2018 and 2019 KABBIs were all Caucasian and heterosexual.

### 3.2. Results

(a)    General Knowledge Self-Rating Scores

Comparisons of pre–post means for general and overall knowledge self-ratings were significant in almost all time periods (Table 2). The post-program scores shown in bold were significant ($p < 0.05$). There were some evaluation periods in which insufficient numbers of participants completed their repeat surveys, and there was one year in which the post-evaluations were inadvertently not sent out due to a change in research coordinators.

(b)    General Knowledge Self-Rating

The maximum total general knowledge self-rating score was 65. Paired data showed that participants consistently rated their knowledge higher across the 13 content areas in the post-assessment, even if these topics had not been highlighted in the activities. The meta-analysis indicated a significant difference in knowledge from the program (Figure 1). There was no heterogeneity, as was expected with the paired samples. Overall, there was an average increase in knowledge of 16.75 (25%).

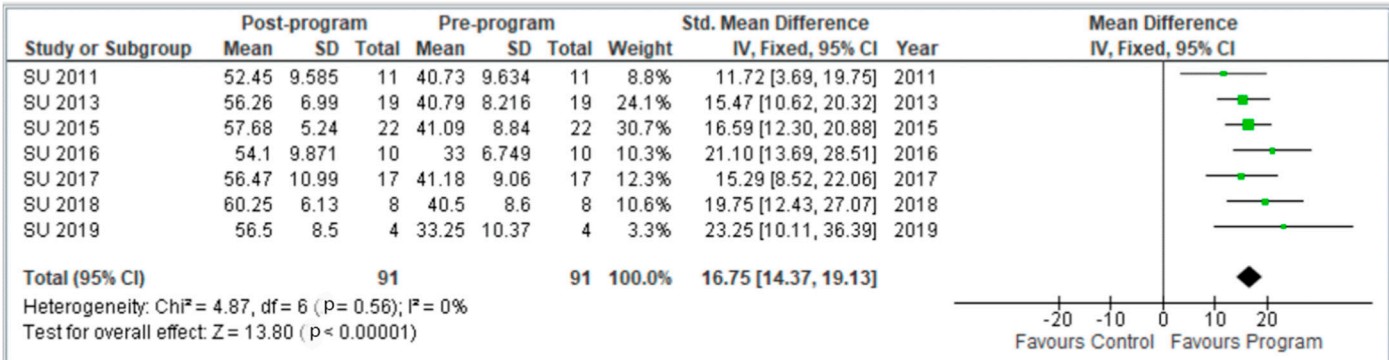

**Figure 1.** General knowledge self-rating scores.

**Table 2.** General knowledge self-rating scores.

| Overall Knowledge Levels | | 2010 | 2013 | 2015 | 2016 | 2017 | 2018 | 2019 |
|---|---|---|---|---|---|---|---|---|
| Healthy relationships | Preprogram | 4.08 | 3.85 | 3.71 | 3.11 | 3.5 | 3.5 | 3.75 |
| | Postprogram | **4.50** | **4.50** | **4.46** | **4.22** | **4.44** | **4.88** | **4.5** |
| | Post 6 month | 4.25 | 4.10 | 4.44 | **3.69** | 4.25 | * | * |
| Warning signs of RV | Preprogram | 3.33 | 3.60 | 3.08 | 2.44 | 3.39 | 3.38 | 3 |
| | Post program | **4.17** | **4.05** | **4.25** | **4.00** | **4.61** | **4.75** | 4.25 |
| | Post 6 month | 3.75 | **3.90** | 4.44 | **4.62** | 4.25 | * | * |
| Interventions with RV | Preprogram | 2.75 | 2.58 | 2.50 | 2.11 | 2.59 | 2.38 | 2.25 |
| | Postprogram | **3.67** | **3.95** | **3.88** | **3.89** | **4.41** | **4.38** | **4** |
| | Post 6 month | 3.50 | **3.90** | **3.89** | **4.00** | **4.13** | * | * |
| Healthy sexual relationships | Preprogram | 3.42 | 3.35 | 3.42 | 3.22 | 3.14 | 3 | 2 |
| | Post program | **4.25** | **4.45** | **4.67** | 4.11 | **4.43** | **4.64** | **4.5** |
| | Post 6 month | 4.00 | **4.40** | 4.33 | **4.46** | **4.28** | * | * |
| Sexual assault myths | Preprogram | 3.25 | 2.85 | 3.04 | 2.56 | 3.17 | 2.74 | 2.25 |
| | Postprogram | **4.00** | **4.40** | **4.52** | **4.22** | 3.25 | **4.75** | 4.25 |
| | Post 6 month | 4.25 | **4.00** | 4.11 | **4.38** | 4.5 | * | * |
| Bystander intervention | Preprogram | 2.50 | 2.30 | 2.42 | 1.78 | 2.56 | 2.63 | 2 |
| | Post program | **4.08** | **3.75** | **4.04** | **3.89** | **4.22** | **4.5** | 4.25 |
| | Post 6 month | 3.75 | **3.70** | 3.89 | **4.08** | 4 | * | * |
| Personal strategies with RV | Preprogram | 2.92 | 2.75 | 2.58 | 2.00 | 2.89 | 2.75 | 1.75 |
| | Postprogram | **3.75** | **4.35** | **4.25** | **4.11** | 4 | **4.63** | **4.75** |
| | Post 6 month | 4.00 | **4.00** | **4.33** | **4.23** | **4.13** | * | * |
| RVresources | Preprogram | 2.75 | 2.45 | 3.04 | 1.78 | 2.56 | 2 | 2 |
| | Post program | **4.00** | **4.45** | **4.54** | **4.00** | **4.28** | **4** | **4** |
| | Post 6 month | 3.75 | **4.20** | 4.11 | **4.15** | 4.25 | * | * |
| Sexual violence resources | Preprogram | 2.45 | 2.45 | 3.08 | 1.78 | 2.88 | 3 | 2 |
| | Postprogram | **3.64** | **4.45** | **4.63** | **4.00** | **4.35** | **4.38** | **3.75** |
| | Post 6 month | **3.75** | **4.00** | 4.11 | **4.23** | **4.38** | * | * |
| Healthy communications | Preprogram | 3.58 | 3.75 | 3.63 | 2.67 | 3.5 | 3.38 | 3.25 |
| | Post program | 4.08 | **4.60** | **4.58** | **4.22** | **4.33** | **4.5** | 4.75 |
| | Post 6 month | 4.25 | 4.30 | 4.33 | **4.46** | **4.13** | * | * |
| Know personal boundaries | Preprogram | 3.25 | 3.70 | 3.50 | 3.00 | 3.22 | 3.5 | 2.75 |
| | Postprogram | **4.33** | **4.55** | **4.29** | **4.44** | **4.33** | **4.75** | **4.75** |
| | Post 6 month | 3.75 | 4.40 | 4.67 | 4.38 | 4.25 | * | * |
| Gender stereotype & RV | Preprogram | 3.33 | 3.50 | 3.63 | 2.78 | 3.22 | 3.5 | 3.25 |
| | Post program | **4.17** | **4.50** | **4.54** | **4.11** | **4.44** | **4.88** | **4.5** |
| | Post 6 month | 4.25 | **4.30** | 4.33 | **4.46** | **4.38** | * | * |
| Media stereotype & RV | Preprogram | 3.67 | 3.30 | 3.92 | 3.00 | 2.88 | 3.38 | 3 |
| | Postprogram | **4.50** | **4.50** | **4.63** | **4.22** | **4.33** | **4.88** | **4.25** |
| | Post 6 month | 4.00 | **4.50** | 4.56 | **4.46** | **4.38** | * | * |
| | n pre | 22 | 37 | 45 | 53 | 33 | 24 | 30 |
| | n (pairs) post1 | 15(11) | 22(19) | 25(22) | 11(10) | 19(17) | 10(7) | 5(4) |
| | n (pairs) 6mos | 9(4) | 12(10) | 9(3) | 19(13) | 12(8) | 2(1 *) | 0 |

Scale 1 (low) to 5 (high); RV = relationship violence. * insuffcent n to compare. **bold** = $p < 0.05$ one sided.

(c)    Module-Specific Knowledge Change

There were 22 questions across the four modules that assessed participants' pre- and post-knowledge of healthy and unhealthy relationships and/or resources. The combined total of their scores by year was used for the meta-analysis (Figure 2). The 2013 group had a markedly different outcome, with a drop in specific knowledge, which was unique to their year. The researchers were unable to identify differences in data entry, demographics, content, or delivery that may have accounted for this. With the omission of the 2013 group, the impact of the program was significant with minimal heterogeneity. Scores increased by an average of 7.84 points post-program.

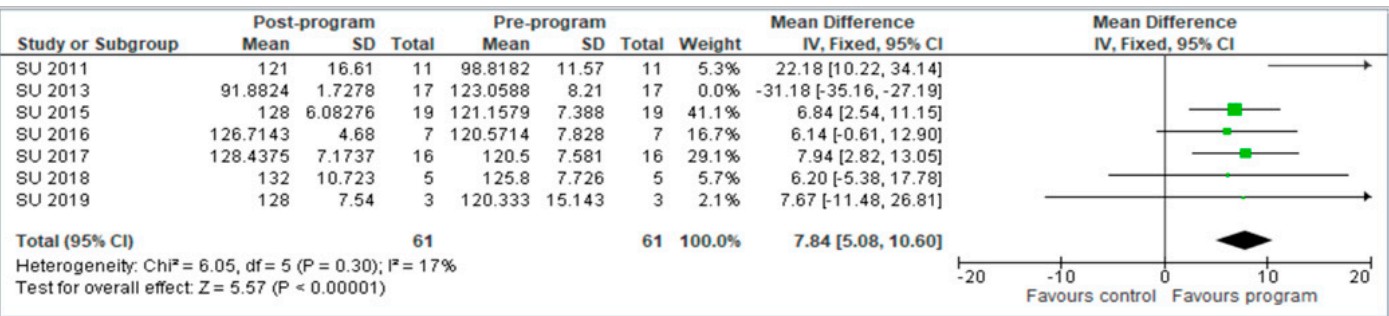

**Figure 2.** Module knowledge questions.

The questions used for the knowledge question score explained 39% of the variability in predicting the knowledge scores using regression analysis. The questions that significantly impacted the score included:

- Gender/media—stereotypes of weak or dependent women could be related to abuse in relationships ($p = 0.024$); abuse between partners could be connected to negative gender stereotypes ($p = 0.022$).
- Sexual relationships—pressuring a partner to have sex is abusive ($p = 0.015$); a partner convincing the other to have sex without contraception is abuse ($p = 0.04$).
- Bystander/boundaries—it is not ok for a partner to go through the other's belongings without permission ($p = 0.005$); partners should not make plans without telling the other ($p = 0.013$).

(d)    Module-Specific Attitude Change

The paired attitude scores were a little more mixed (Figure 2). Again, there was a reverse impact of the 2013 data for unknown reasons. The 2013 data were not included in the meta-analysis, as the differences could not be estimated. The overall effect was smaller (2.62) than that for knowledge and behavioural intents, and there was greater variability in the confidence intervals. Overall, the results indicated that the program was significantly associated with a change in attitudes with minimal heterogeneity (Figure 3). A total of 28 attitude questions across the modules were used to create the final score. There were some years with mixed results, but most were not heavily weighted in the analysis due to sample size.

The regression analysis showed that the questions used explained 89% of the variability in attitude scores. There were five questions that significantly contributed to the score:

- Healthy relationships—verbally putting down a partner is a form of abuse ($p = 0.006$); peers can play a big role in stopping abuse in their friends' relationships ($p = 0.42$).
- Sexual relationships—a person should not touch their partner in a sexual way unless they want to be touched ($p = 0.01$); if someone is very drunk or under the influence of drugs and their partner has sex with them, it is sexual assault ($p = 0.002$).
- Boundaries/Bystanders—someone should only intervene in other people's relationships if the situation is dangerous ($p = 0.007$).

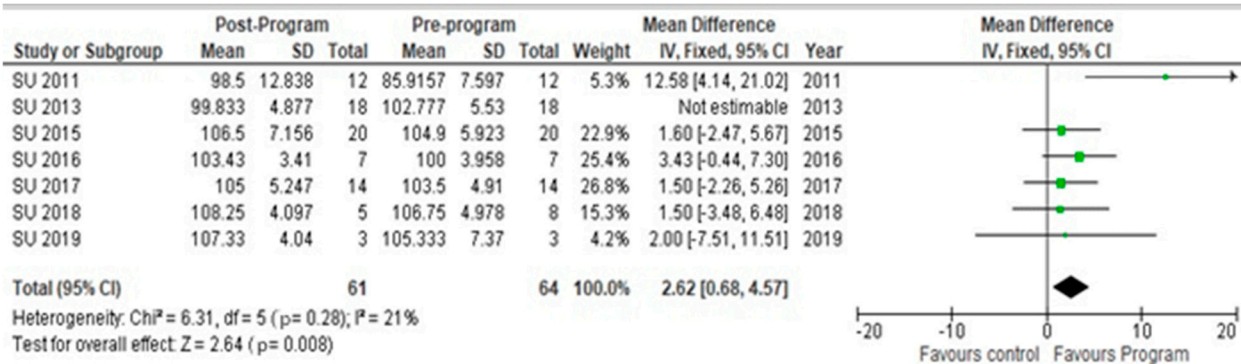

**Figure 3.** Attitude changes.

(e)     Behavioural Change

There were 21 questions related to behavioural intents. The paired results were significant, and, again, there was no heterogeneity detected (Figure 4). There were two years in which there were insufficient data for analysis. Similar to the attitude results, there are some years in which the confidence interval ranged from significant to non-significant, but these were not heavily weighted, mostly due to sample size. Overall, however, the effect was statistically significant, and there was no heterogeneity.

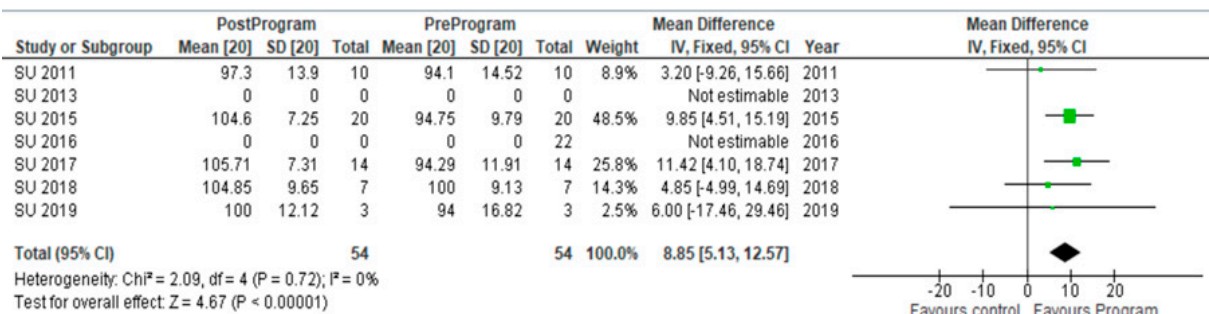

**Figure 4.** Module behavioural intents.

Regression analysis showed that 99% of the variance in behavioural intent change was explained by 20 of 21 of the predictors. All were significant (<0.01) with the exception of the intent to break up if regularly insulted by their partner.

## 4. Discussion

There was a higher rate of participants who had witnessed relationship violence at home than is seen in the literature. This may be due to differences in definitions between studies, but it could also be a result of self-selection. Those who have witnessed relationship violence in childhood are a higher-risk population, however, and attracting them to a prevention program is important to prevent continued violence in their lives. It may also reflect a lack of other relevant resources at the university level.

Overall, both the yearly *t*-tests previously conducted and the meta-analysis support the effectiveness of the program in improving knowledge, attitudes, and behavioural intents in this relatively high-risk population. Changes in attitudes were less profound compared with behavioural intents and knowledge. Attitudes are generally harder to change than knowledge, so it may be that the sessions were not long enough or that the content was not focused enough to create change. Alternatively, the change in knowledge may result in some unsettling of ideas and uncertainty. The *t*-test data in the yearly analyses showed stronger knowledge after six months in many areas, perhaps as a function of the additional activity and perhaps opportunities to reflect or become more aware of these issues with other activities on campus. Unfortunately, we did not have enough paired samples to

make a six-month analysis meaningful. The changes in knowledge and attitudes were consistent with the systematic reviews conducted previously (Wong et al. 2021). It was noted that data from women and Caucasian populations may overestimate the effectiveness of knowledge changes (Wong et al. 2021), which needs to be considered here as well. The consistent increases across years, especially in knowledge and behavioural intent, lend some confidence to the results, however.

The significant changes suggest sufficient power and effect size to detect a difference. There is a potential for type I error due to selection bias and effects of attrition, although the meta-analyses weighted the larger sample sizes more heavily. The repeated significance of the program across differences in students and topics provides some confidence that the program is effective. Our student participants are becoming more diverse in ethnicity as well as gender identity and sexual orientation, and yet the program continues to show effectiveness. Another measure of effectiveness was the enthusiasm of peer facilitators and participants, with many of them returning as peer facilitators the next year and some changing the focus of their careers to work with vulnerable populations. One of the recommendations for effective prevention programs by Crooks et al. (2019) was to be flexible in programming to address cultural and ethnic diversity, tailoring content to the needs of participants. Although our participants were still predominantly Caucasian and heterosexual, there was diversity in both participants and peer facilitators. The re-evaluation of content for modules each year allowed for flexibility in meeting the needs of more diverse populations. Lacking each year, however, was the number of male participants. Each year, efforts have been made to recruit and attract men, but these efforts have had limited success.

There were concerns with the 2013 data, with reverse effects of the program on attitudes and module-specific knowledge. It is unknown why that program differed so markedly from the others. Data were double-checked and accurate. Anecdotally, it was noted that sometimes, the initial post-evaluation created some discomfort and drops in scores for some participants as they absorbed new information or were introduced to new concepts, and then the six-month scores increased. It is possible that either the mix of students or activities introduced created this unrest or uncertainty. The low response rate to the six-month questionnaires prevented in-depth analysis of those effects.

It is difficult to know what specific aspect of the program contributed to its effectiveness. The topics, format, exposures, peer influence, and researcher or community support were all integral to the program. Crooks et al. (2019) suggested that comprehensive programs have demonstrated value, with potential benefits from a broader focus rather than on any one specific aspect of violence prevention (e.g., sexual assault or bystander intervention). It was interesting to note that the specific topics within each module changed slightly from year to year, depending on the current issues, and many of the topics on the KABBI were never formally discussed or included. Despite this, there were knowledge increases in every area. There were many opportunities for student informal discussions, perhaps creating space for this knowledge or awareness. In addition, the KABBI and the DRS may act as an intervention. We conducted a factor analysis on the KABBI tool at one point, hoping to shorten the number of questions. Students anecdotally described learning from the KABBI and DRS tools and reflecting on or discussing issues as a result. As a result, the decision was made to keep the full KABBI, as the questions provided opportunities to raise awareness and perhaps stimulate reflection on topics the students may not have considered. The learning across issues with our program supports the benefits of a broad program identified by Crooks et al. (2019).

A surprising finding was the limited baseline knowledge about how to intervene if dating violence is witnessed or experienced, personal strategies for prevention, and community resources for either dating violence or sexual violence. This was consistent across the decade, despite increasing initiatives across the university to raise awareness. Examples included the almost yearly community fairs, a yearly interdisciplinary violence prevention forum, advocate training for residence advisors, and appointing a coordinator

for dating, domestic, and sexual violence on campus. This emphasises the need for programs to include bystander skills, as noted in all three systematic reviews (Crooks et al. 2019; Finnie et al. 2022; Wong et al. 2021). Students continue to enter the university with very low levels of knowledge about what to do, and yet peers are a key factor in victims of violence seeking help (Carter-Snell 2007; Padilla-Medina et al. 2022). If the peers do not know where to go or how to help, then the risk of negative consequences increases.

It is of concern that there were not strong effects and that sometimes there were reverse effects with some of the specific content areas on KABBI. For instance, many students still believed that men were better leaders and should make more decisions in a family, and many were not aware of some of the sexual assault myths or the significance of prior violence or experiencing sexual assault in a relationship as danger signs.

Another limitation was the limited six-month follow. As noted in the systematic reviews, behavioural skills tend to decrease over time (Finnie et al. 2022; Wong et al. 2021). Unpaired data showed persistent increases in knowledge, including in bystander skills, after six months. There were insufficient paired data for analysis or meaningful interpretation, but the results were quite consistently positive (Table 2). Future efforts are needed to improve response rates to assess the long-term impact on knowledge, attitudes, and behavioural intents. Recommendations for future programs would be to work with peer facilitators to ensure that key knowledge content is addressed through discussion or activities. Examples would be recognising that the greatest predictor of violence is a history of prior violence and addressing gender myths. There is still a need, however, to be flexible with the content and to allow modification for changing student populations. Recruitment of men, gender diverse, and different cultural groups should be encouraged, along with increased attempts to improve six-month response rates.

**Author Contributions:** Conceptualization, C.J.C.-S. and D.G.W.; methodology, C.J.C.-S. and D.G.W.; Software, C.J.C.-S.; validation C.J.C.-S. and D.G.W.; investigation C.J.C.-S. and D.G.W.; resources, D.G.W.; data curation, C.J.C.-S.; writing, C.J.C.-S. and D.G.W.; editing, C.J.C.-S.; supervision, D.G.W.; project administration, D.G.W.; funding acquisition, D.G.W. All authors have read and agreed to the published version of the manuscript.

**Funding:** This research was funded by the Government of Alberta, grant # 09528054 Th.

**Institutional Review Board Statement:** The study was conducted in accordance with the Declaration of Helsinki, and approved by the Institutional Review Board for Ethics of Mount Royal University, # 100933, 2010–2024.

**Informed Consent Statement:** Informed consent was obtained from all participants in the study. Participants remained anonymous in all data collection.

**Data Availability Statement:** The data presented in this study are available on request from the corresponding author. The data are not publicly available due to university ethics restrictions done.

**Acknowledgments:** The authors wish to thank the following people for their considerable contributions to the project: Patricia Kostouros, Leslie Tutty, Anne Troy and Christine Brownell for their contributions and vision to developing the program and for co-facilitating the program as faculty advisors; as well as the the project coordinators, peer facilitators and the many student participants.

**Conflicts of Interest:** The authors declare no conflict of interest.

## Appendix A. KABBI Questionnaire

*Appendix A.1. Knowledge Self-Rating Questions*

Please use the scale 1–5 to rate your knowledge about the following topics:
(1 = very limited, 2 = somewhat limited, 3 = adequate, 4 = somewhat well informed, 5 = extremely well informed)

1. Healthy vs. unhealthy dating relationships.
2. Warning signs of dating violence.
3. Interventions to reduce dating violence.

4. Healthy versus unhealthy sexual relationships.
5. Sexual violence myths and stereotypes.
6. How to intervene if witness/learn of dating violence.
7. Personal strategies to reduce dating violence.
8. Community resources to assist with dating violence.
9. Community resources to assist with sexual violence.
10. Healthy versus unhealthy communication styles.
11. Personal boundaries and boundary setting.
12. Gender Stereotypes and role in violence.
13. Media stereotypes and role in violence.

For all following questions, please use the 1–5 scale as shown:
(= completely disagree, 2 = somewhat disagree, 3 = neutral, 4 = somewhat agree, 5 = completely agree)

Appendix A.1.1. Module: Healthy Relationships

(a) Attitudes

1. There is nothing wrong if a person wants to spend time away from their partner.
2. A person does not have the right to be physically violent (e.g., hit or push) if they are being insulted.
3. A person is responsible for what they do when they are drunk or using drugs.
4. A person who loves their partner should not have to be willing to do anything to keep them happy.
5. Peers can play a big role in stopping abuse in their friends' relationships.
6. Being sworn at is no worse for women than for men.

(b) Behaviours

1. I would tell someone I trusted if I was being abused by my partner.
2. I would tell someone I trust if I was abusing my partner.
3. I would encourage a friend who is being abused to tell a resources person (e.g., parent, teacher, health care worker, counsellor, community agency).
4. I would tell a resource person if a friend were being abused.
5. I would break up with a partner if they insulted me regularly.
6. I would break up with a partner if they pushed or shoved me regularly.
7. I would help a person who is being hit by their partner.
8. It's not OK to hit my boyfriend even if they do something to deserve it.
9. I can resist hitting my partner if they make me angry.

(c) Knowledge

1. If a person never lets their partner out of their sight this is a sign of abuse.
2. If a person has hit a previous partner, they are more likely to hit their current partner.
3. Wanting to be a partner's only friend is a warning sign that the relationship may be abusive.
4. Telling a partner who they can spend time with is a form of abuse.
5. Feeling angry at your partner is not a form of abuse.
6. Putting down a partner is a form of abuse.
7. Pushing a partner is a form of abuse.
8. Insisting that a partner wears certain clothes is a form of abuse.
9. If a partner cries and apologises after hitting they are still likely to hit again.

Appendix A.1.2. Module: Gender and Media

(a) Knowledge

1. Stereotypes about how strong or macho guys should be related to abuse in relationships.
2. Stereotypes about how weak or dependent girls should be related to abuse in relationships.
3. Abuse between partners is connected to gender stereotypes.
4. Music lyrics affect how people think about men and women.
5. The media can influence people's ideas about violence.

(b) Attitudes

1. Using swear words is no worse for women than for men.
2. Men should not make more decisions in a family than women.
3. Men are not better leaders than women.
4. Women are not weaker than men and don't need to be looked after.

(c) Behaviours

1. I would be able to take action so that the university does not hold activities that promote violence.
2. I would be able to take action to have gender stereotype messages removed from the university.

Appendix A.1.3. Sexual Relationships

(a) Attitudes

1. If the two people have been dating a long time it is still not ok to pressure the other to have sex.
2. It is not all right to joke with others about a partner's sexual performance.
3. A person should not touch their partner in a sexual way unless they want to be touched.
4. When someone pays on a date it is still not ok to pressure the other partner to have sex.
5. When girls say "no" they don't sometimes really mean "yes".
6. Some men may say "no" as they are not always willing to have sex.
7. If someone is dressed in sexy clothing it is not their fault if their date forces them to have sex.
8. Men do not have stronger biological urges than women so can resist forcing themselves on women.
9. If a person goes home with a date and is forced to have sex it is not their own fault.

(b) Knowledge

1. A man can be sexually assaulted by a woman.
2. A man can be sexually assaulted by another man.
3. If someone is very drunk or under the influence of drugs and their partner has sex with them, it is sexual assault.
4. Pressuring a partner to have sex is abusive.
5. If a partner convinces the other to have sex without pregnancy protection, it is abuse.
6. If someone is drunk or under the influence of drugs, they are more vulnerable to sexual assault.
7. Someone who has sex with their partner when they know the other doesn't want to have sex is abusive.
8. Comments or jokes about someone's sexual performance are abusive.
9. Forced sex is a sign of increasing danger in a relationship.
10. Sexual assault is not about sex, it is about power and control.
11. Sexual violence in a relationship is a warning sign that personal safety is at risk.

    12.    I would know where to get help if my partner forced me to have sex.

(c)    Behavioural Intent

    1.    I would break up with a partner if they regularly insisted on sex when I didn't want it.

    2.    I would do something to help a person who is being forced to have sex.

    3.    I would stop a friend from pressuring their partner to have sex.

    4.    I would say something if I saw someone trying to take advantage of another's drunken state to have sex.

    5.    If I saw someone was at risk for being sexually assaulted, I would not be hesitant to help, even if was unsure if others present would support me.

Appendix A.1.4. Module: Boundaries and Bystanders

(a)    Knowledge

    1.    If someone regularly cuts off their partner when they are giving their opinion it is abusive.

    2.    If I know someone is being abused in their relationship I would know where to send them for help.

(b)    Attitudes

    1.    It is not ok for a partner to go through their partner's belongings without the other's consent.

    2.    Partners should not make plans without telling the other.

    3.    It is not ok if a partner does not listen to the other's views.

    4.    Someone should intervene in other people's relationships if the situation is dangerous.

(c)    Behavioural Intents

    1.    I would be able to state my concerns about a posting on social media that makes fun of someone's partner, or which is abusive.

    2.    I would be able to state my concerns about my relationship with my partner.

    3.    I would be able to step in if a friend was arguing with their partner and their behaviour became aggressive.

    4.    I would be able to tell someone if I was being abused.

    5.    If I saw someone acting aggressively toward their partner, I would not be hesitant to help them if I was unsure if others present would support me.

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
