# Peer review of "“Stepping Up”: A Decade of Relationship Violence Prevention"

_socsci, doi:10.3390/socsci12090501_

Round 1

Reviewer 1 Report

The authors deal with an interesting and important topic, focusing on anti-violence programs and the characteristics that determine their effectiveness.

However, in my opinion, the article needs some improvements to be accepted.

Here are my suggestions for authors:

INTRODUCTION:

1) The introduction about the diffusion of the problem should be extended. Why did the authors choose to speak only of North America (also considering that the study is carried out in Canada)? Is there a greater concentration of the phenomenon there than in other countries? What does the literature say? A comparison using international data would be helpful.

2) The exposure about the types of intervention programs should be extended. The authors talk directly about the programs in Canada, briefly touching on them and immediately alluding to the "missing programs".

3) When the authors write "In 2010 our team developed Stepping Up based on Making Wave" are they already talking about the current study? Why already at this point of the introduction? Authors should restructure the text better, making it clearer and more coherent.

METHODS:

4) The "Studies and Participants" paragraph is also confusing. Numerical information on the participants in the various phases is completely missing. And the order of the phases is not clear.

5) The paragraph "Measurement Instruments" lacks information about the reliability of the scales used.

RESULTS:

6) The paragraph "Participants" is currently not very useful: it adds nothing more than the previous one on the participants and only refers the reader to the table. Authors should report the most relevant descriptive statistics to summarize the sample (if they want to keep the paragraph).

7) The formatting of Tables and Figures needs to be improved, they are difficult to read.

DISCUSSION

8) The authors should clarify better what they believe is the greatest result obtained, in the light of the objective they have set. What are the characteristics that, according to them, make an intervention effective? The discussion is dispersive and hasty.

9) The discussions lack a comparison between the results emerging from the study and the national and international literature on the subject. Authors should compare their findings with reports from articles that have investigated anti-violence programs.

Some examples could be:

Esposito, C., Di Napoli, I., Esposito, C., Carnevale, S., & Arcidiacono, C. (2020). Violence against women: A not in my back yard (NIMBY) phenomenon. Violence and gender, 7(4), 150-157.

Shorey, R. C., Zucosky, H., Brasfield, H., Febres, J., Cornelius, T. L., Sage, C., & Stuart, G. L. (2012). Dating violence prevention programming: Directions for future interventions. Aggression and violent behavior, 17(4), 289-296.

Close, S.M. (2005). Dating violence prevention in middle school and high school youth. Journal of Child and Adolescent Psychiatric Nursing, 18(1), 2-9.

Foshee, V. A., Bauman, K. E., Arriaga, X. B., Helms, R. W., Koch, G. G., & Linder, G. F. (1998). An evaluation of Safe Dates, an adolescent dating violence prevention program. American journal of public health, 88(1), 45-50.

Avery-Leaf, S., Cascardi, M., O'leary, K. D., & Cano, A. (1997). Efficacy of a dating violence prevention program on attitudes justifying aggression. Journal of adolescent health, 21(1), 11-17.

REFERENCES:

10) The authors should definitely increase the study of the literature on the investigated theme. The list of references is visually limited.

Author Response

Thank you for your comments. We have reworked the introduction to be more clear overall. Since our students range from never having dated to being married or divorced there is spousal data that is relevant. This may be confusing so we changed the title to the increasingly common term "relationship violence" and defined it in the introduction. The process for the environmental scan is more fully explained (https://bmjopen.bmj.com/content/9/9/e029805). The discussion of type errors was removed and simplified but a discussion of the single study limitations has to remain as that is the reason why this study was conducted- to pool the data.  Corrected the formatting and numbering for the tables.  Supported the discussion with the three most relevant systematic reviews in the prevention literature relevant to young adults/university and builds on their findings. revised the conclusion accordingly.

Reviewer 2 Report

Overall, this manuscript was poorly written and confusing; there were numerous grammar issues, which made it difficult to follow what your study was about. The writing needs to be dramatically changed to be easier to comprehend for readers. Introduction needs to be re-organized to flow better; only provide context about what your study is about; not spousal violence, as you used HS data. What is a dose relationship? What is an environmental scan? Confusing to see all your Type errors in the literature review; should be moved to your Methods about the program overall. Are you analyzing your own data or doing a meta-analysis of all of your data? Very confusing. Participants section should be in Methods, not results. All of the table information should explained in Methods/Measures. More context should be provided in the background/lit review. Tables are not formatted correctly (APA). Discussion is not appropriate, and reads more like opinions, rather than connecting back to the literature and how your study builds upon previous findings. Final conclusion is poorly written.

Quality of English is fair, but overall, there were so many grammar issues (missing words; poorly written; passive writing) that it was difficult to follow.

Author Response

Thank you for your comments. We developed the program prior to the available systematic reviews on dating violence prevention which is why we did not initially reference it. We have incorporated that now both in the introduction and in the discussion of findings. 

Reviewer 3 Report

Something important is unclear from the description under methods.  Between 2011 and 2019, there were a number of "participants" as identifed in table 2, sample sizes.  Do these figures (and therefore the results) refer to the "peer educators" only?  Or, are these only students who participated in the workshops?  Or are they both?  This is of course important for both analysis and interpretation purposes, but it is unclear to me who exactly the "participants" are.  

I find it notable that between 44.8% and 60% of participants witnessed IPV at home as a child. How was this defined in the question?  I find it notable that relatively large percentages of participants have experienced domestic violence directly.  I am unfamiliar with the prevalence values for these measures in the context.  Are these higher, lower or virtually the same as prevalence values in college-age individuals at large?  Is it possible that their experiences with domestic violence (directly) or IPV (witnessed) that would make them pre-disposed to be participants and if so, in what ways might this impact results or be important to the interpretation/discussion part of the paper.

The paper would be strengthened with a clearer Data Analysis description. Analysis of differences in participants (or lack thereof) is not presented in sufficient detail.  We see Table 2 but we don't know if the differences in, for example the female/male ratio from year to year is significant.  What does it mean for the effect results that the participants were predominantly female? Did the analysis (the results of which presented only focus on the question of components) control for background characteristics?  If you run the analysis separately for males and females, do the results differ? Whether this is not part of the analysis for whatever the authors' reasons: not of interest, or beyond the scope of the study, whatever the reason, it's important to explain.

It's helpful that the authors note:  What was interesting is that the specific topics within each module changed slightly from year to year, depending on the current issues, and many of the topics on the KABBI were never formally discussed or included. Lines 284 and 285), but it begs the question of how much of the module was actually the same - how are we assured that enough of it was consistent to actually be able to reliably pool the scores as if the intervention is unform from year to year?  

Recommendations from line 319 onward should be elaborated.

There are some very minor language issues that a proofer can pick up and address.  Other than that, no problem with language.

Author Response

Thank you for your comments. We have revised the order of tables. The small size/attrition of participants is a concern as noted but we have added more discussion as to comparisons of the subsample with larger sample and, in the discussion, added comments on power and type I error.  The discussion has been revised quite a bit to show how our results fit with the larger body of research and implications for future research.  We have included a discussion of why the 2013 data may be different and why they were not included in the attitude analysis.

Reviewer 4 Report

The order of the tables is incorrect.

The major limitation of this study is the large loss of participants. This fact is almost impossible to overcome.

The authors should improve the discussion. The discussion must justify its results based on scientific evidence. In fact, only two references appear in the discussion. It is necessary to give an explanation about the year 2013, for example. Major changes are necessary.

Author Response

Thank you for your feedback. We have clarified throughout the revisions that only participant data are included, since peer facilitator data has been examined in previous publications.  We also addressed the high rate of IPV in our population. It is high in our university overall but those with experiences of violence are also more likely to seek out this type of program.  They are in need of it most. The high prevalence of witnessed IPV is well recognized in the literature as a risk for experiencing IPV themselves which is why the program is needed.  The data analysis sexction includes a discusison now of how the subsample is similar to the larger sample despite attrition.  It is noted in the discussion that female, caucasian populations may overestimate effect (in our revised version). There were not enough males to do any sub-analyses and that is a limitation of the program we noted.  We added a discussion of how the modules are still similar despite peer facilitators choosing different activities and specific topics and improved the rationale as to why we could pool those data.  Expanded the recommendations as noted.

Round 2

Reviewer 1 Report

In my opinion the paper can be accepted.

Reviewer 2 Report

Authors did a nice job of revising the previous version.

Reviewer 4 Report

x